# Narrative Integration: An In-Depth Exploration of the "Buddha Story Stele" in the Maiji Mountain Grottoes

**Zejie Lin** [1], **Zhijun Li** [2] **and Meizi Xie** [1,*]

1    School of Humanities, Guangzhou University, Guangzhou 510006, China; lzj@gzhu.edu.cn
2    School of History and Culture, Henan University, Kaifeng 475001, China; hndxllizhijun@163.com
*    Correspondence: meizi.xie@gzhu.edu.cn

**Abstract:** This research delves into the intricacies of the "Buddha Story Stele" in Cave 133 of the Maiji Mountain Grottoes, China, examining the sculptural combinations and conceptual nuances rooted in Buddhist culture from the 5th to the 6th centuries CE. The research focuses on discerning the identities of the "Two Adjacent-Seated Buddhas" and the Cross-Legged Bodhisattva carved on the stele, concurrently delving into the embedded symbolic significance within its structural composition. Our investigation posits that the upper, middle, and lower segments of the "Buddha Story Stele" respectively symbolize the post-Nirvana Dharmakāya Shakyamuni, the Bodhisattva Shakyamuni, and the Buddha Shakyamuni of Sumedha. Advancing scholarly discourse, it reevaluates the Cross-Legged Bodhisattva's identity and the configuration of the "Two Adjacent-Seated Buddhas", elucidating the interplay of imagery and conceptual themes. This study provides pivotal insights into the sculptural arrangement and religious thought transmission in the Maiji Mountain Grottoes, contributing significant academic and cultural value to preserve this unique heritage.

**Keywords:** religious iconography; Chinese Buddhist art; cave art interpretation; Two Adjacent-Seated Buddhas; Cross-Legged Bodhisattva; Maiji Mountain Grottoes





## 1. Introduction

The various inscribed iconic steles found in Cave 133 of Maiji Mountain not only divulge information about the widespread religious trends during the 5th to 6th centuries in Central Asia and various parts of China, such as the veneration of the *Lotus Sūtra* and Shakyamuni Buddha, but also unveil a distinctive and regional development shaped by the amalgamation of Central Asian culture and the unique material culture of the Maiji Mountain area during the medieval period. In particular, the "Buddha Story Stele" in Cave 133 (commonly known as Stele No. 10 in Cave 133) is divided into three distinct segments, featuring the "Two Adjacent-Seated Buddhas", the Cross-Legged Bodhisattva, and the Seated Preaching Buddha, with depictions of Buddha stories and Jataka tales flanking both sides (Figure 1).

This unconventional layout breaks free from traditional Indian paradigms like the Gandharan, Mathuran, Gupta, and South Indian styles. Instead, it represents an alternative form of regional development in Buddhist beliefs around Maiji Mountain. This stele extends from being merely a static record of traditional religious art; it serves as a temporal conduit, transporting us to the live intersection of the religious belief and cultural amalgamation of this era. Through a meticulous examination of this stone monument, we intend to delve deeply into the diversity of religious beliefs, the mutual influence of cultures during this historical period, and the unique role played by the "Buddha Story Stele" within this distinctive historical context.

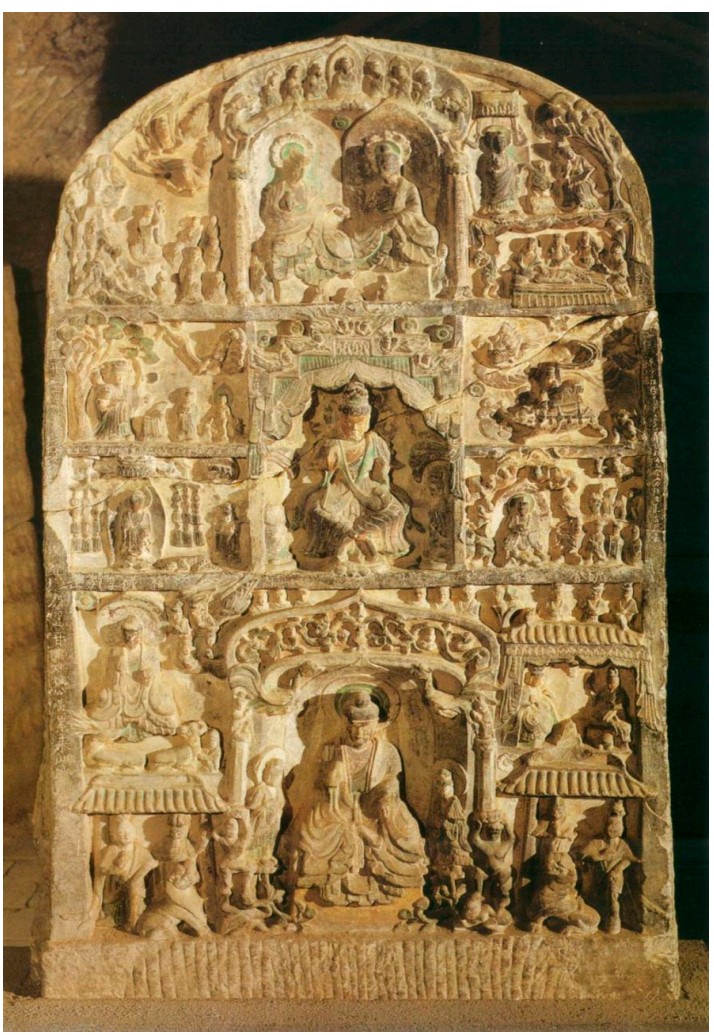

**Figure 1.** The "Buddha Story Stele" from Cave 133 of Maiji Mountain, Northern Wei, preserved in the Gansu Provincial Museum in China.

Presently, scholarly research on this iconic stele primarily revolves around the identification of the Buddha stories and Jataka tales and the exploration of its artistic style (Zhang and He 1995; Xiang 1998; Chen 2004; Wang 2005, pp. 49–52; Y. Zhang 2017; Yang 2019; Sun 2021). Notable contributors to this field include Yiming Jiang (1990), Zhong Long (2017), Wong (2004, pp. 127–30), and those who delve into the artistic attributes of the stele. In terms of understanding the interrelation of the diverse images on the stele, some scholars, drawing from their interpretations of the imagery, posit that Stele No. 10 stands out as an exceptional work from the first half of the 6th century. They argue that it adeptly merges depictions of the Trikalea Buddhas' teaching and Buddha stories, which have been present since the era of the Northern Wei dynasty (386–534 CE) (Chen 2004, p. 142). Conversely, other scholars, from the vantage point of the *Lotus Sūtra*, provide an alternative perspective. Some scholars offer an interpretation from the perspective of the *Lotus Sūtra*, suggesting that the depictions of the "Two Adjacent-Seated Buddhas", Cross-Legged Bodhisattva, and Seated Preaching Buddha symbolize various aspects of the *Lotus Sūtra*. These include the *Lotus Sūtra* itself, the Pure Land of Tuṣita, and the founder of the *Lotus Sūtra*. Along with depictions of stories representing the Buddha's skillful means of teaching on both sides, and representations of both Mahayana and Hinayana teachings, such as the Initial Turning of the Dharma Wheel and Manjushrī, all of them enhance our understanding of the faith in the *Lotus Sūtra* during the Northern Wei dynasty (Li 2008). Lee (2010, pp. 61–67) interprets the design and placement of the Nirvāṇa image on the stele,

emphasizing the explicit separation of the Nirvāṇa image from scenes related to the biographical cluster, with a particular focus on depicting the story of Dīpamkara's prophecy of the future Buddha.

Nonetheless, some academic controversies persist, necessitating further examination. Firstly, the identity of the Cross-Legged Bodhisattva is multifaceted, and the presence of Jataka tales and Buddha stories centered around Maitreya seems to contradict the primary focus on Shakyamuni's veneration. Moreover, there is an ongoing debate in the academic community concerning the roles of Shakyamuni and Prabhūtaratna in the "Two Adjacent-Seated Buddhas" within the Trikalea Buddha,[1] and they should be viewed as a combined representation, merging Shakyamuni and Prabhūtaratna into a singular entity from a doctrinal standpoint. Lastly, while recent scholarship has examined the overall imagery of the stele, the connections between the central figures and the surrounding Buddha stories, as well as the Jataka tales, clearly partitioned into three distinct sections, have yet to receive comprehensive discussion. Consequently, building upon prior iconographic content identification, our study systematically redefines the identities of the significant figures, including the "Two Adjacent-Seated Buddhas" and the Cross-Legged Bodhisattva. This redefinition is achieved through a meticulous analysis of the relationships between each central icon and the surrounding narrative scenes. Furthermore, our research methodically elucidates the intricate compositional relationships and nuanced conceptual underpinnings inherent within the entire sculptural ensemble of the stele.

This study holds cultural significance by advancing the understanding of the holistic sculptural arrangement and the nuanced transmission of religious thought in the Maiji Mountain Grottoes. It not only contributes to refining our comprehensive understanding of this archaeological site but also provides substantive insights into the noteworthy role played by Buddhist art in the cultural development of the Central Plains during the Northern Wei dynasty. The introduction of a novel academic perspective is poised to stimulate broader research interests, thereby catalyzing a more profound exploration of ancient Chinese Buddhist art and culture.

## 2. Interpreting the Nature of Dharmakāya through the "Two Adjacent-Seated Buddhas"

The "Two Adjacent-Seated Buddhas", Shakyamuni and Prabhūtaratna, are mentioned in Chapter 11: "The Sight of the Treasure Stupa" of the *Lotus Sūtra*.[2] According to the scripture, Prabhūtaratna once made a grand vow to propagate the teachings of the *Lotus Sūtra*. As a result of this vow, whenever someone expounded the *Lotus Sūtra*, the Treasure Stupa of Prabhūtaratna would manifest. This manifestation both signifies blessings and approval for the speaker and confirms the profound authenticity of the *Lotus Sūtra* (Lai 1981, p. 460; *Miaofa lianhua jing*, pp. 32–33; Strong 2004, pp. 36–39).

### 2.1. Revisiting the Theme of "Two Adjacent-Seated Buddhas"

The imagery of the "Two Buddhas Seated Side by Side" is derived from Chapter 11 of the *Lotus Sūtra*. The dual Buddhas symbolize the concept that a Buddha from a distant past manifests in the present and future whenever the *Lotus Sūtra* is expounded (Rhie 2010, p. 137). This artistic representation is not found in India or Central Asia, making it a unique creation in the Chinese art of the *Lotus Sūtra* (Y. Zhang 2017, p. 13). From the existing examples, the earliest known icon of "Two Adjacent-Seated Buddhas" in China can be traced back to the mural at the 11th niche of Cave 169 in Bingling Temple, dating to the first year of the Jianhong era in the Western Qin regime (420 CE).[3] Subsequently, similar depictions have emerged in various locations, including the Yungang Grottoes, Longmen Grottoes, and Dunhuang Grottoes.

2.1.1. The Development and Transformation of "Two Adjacent-Seated Buddhas"

Y. Zhang (2017, p. 13) pointed out that "during the Northern Wei period when the artistic form of the *Lotus Sūtra* Transformation had not yet appeared, the 'Two Adjacent-

Seated Buddhas' in the Dunhuang Grottoes was used as a symbol of *Lotus Sūtra* faith and representative of *Lotus Sūtra* art". However, in the Dunhuang region, the independent "Two Adjacent-Seated Buddhas" icon did not fade into history with the emergence of the mature *Lotus Sūtra* transformation. Instead, they continued to play a significant role with their unique meaning in combination with other icons. Therefore, based on the authors' observation, the development of the "Two Adjacent-Seated Buddhas" within the Buddhist iconic system generally progressed in two directions.

Firstly, as an early representation of the faith in the *Lotus Sūtra*, the "Two Adjacent-Seated Buddhas" gradually evolved into the central theme of the *Lotus Sūtra* transformation. In Cave 331 of the early Tang period at the Mogao Grottoes, a vivid portrayal of the *Lotus Sūtra* transformation unfolds with the "Two Adjacent-Seated Buddhas" at its core, symmetrically surrounded by Buddhas from the ten directions, along with figures like Manjushrī, Samantabhadra, and Kṣitigarbha Bodhisattva (Figure 2). In what is known as the "Grotto of *Lotus Sūtra*" (Grottoes No. 23) from the flourishing Tang dynasty (618–907 CE), the "Ceremony in the Air" represented by the "Two Adjacent-Seated Buddhas" and the "Assembly on the Vulture Peak" represented by the image of preaching on the Vulture Peak are both found in equivalent compositions on the respective south and north walls of the grotto (Figures 3 and 4). This arrangement underscores the heightened significance of the Spirit Vulture Peak preaching event during this period. Subsequently, the "Assembly on the Vulture Peak" gradually became the central theme of the *Lotus Sūtra* transformation, while the proportional space dedicated to the "Two Adjacent-Seated Buddhas" gradually diminished (Figure 5).

Secondly, the "Two Adjacent-Seated Buddhas", with their unique symbolism, have been handed down as separate revered icons. Beyond the cave icons, a significant number of standalone gold, bronze, and stone "Two Adjacent-Seated Buddhas" icons have been well preserved (Y. Zhang 2017, pp. 50–56), distinctly illustrating their tendency to be regarded and worshipped as separate revered icons.

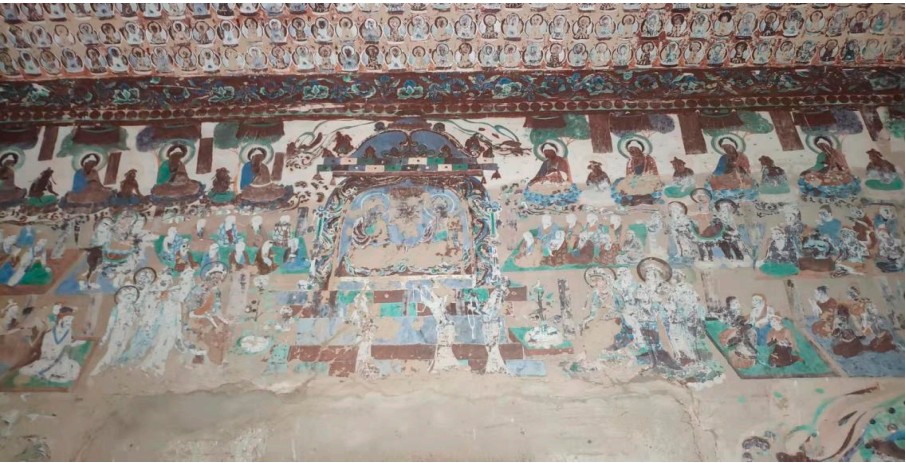

**Figure 2.** The east wall vault of Cave 331 in the early Tang dynasty (618–750 CE) at Mogao Grottoes, with "Two Adjacent-Seated Buddhas" as the central theme in the transformation of the *Lotus Sūtra*.

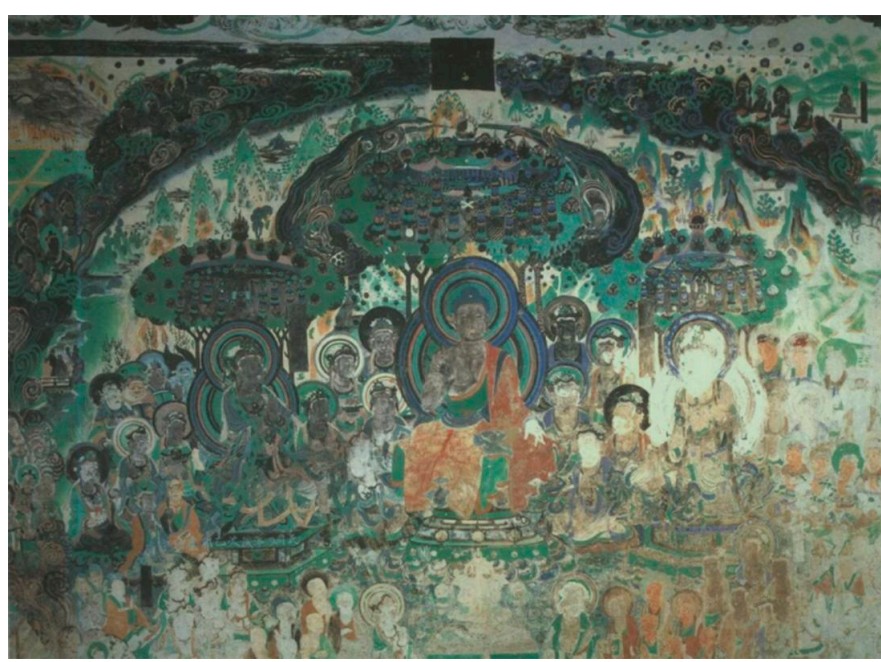

**Figure 3.** The "Assembly on Vulture Peak", transformation of the *Lotus Sūtra* on the Northern Wall in Cave 23 during the flourishing Tang dynasty (704–780 CE) at Mogao Grottoes.

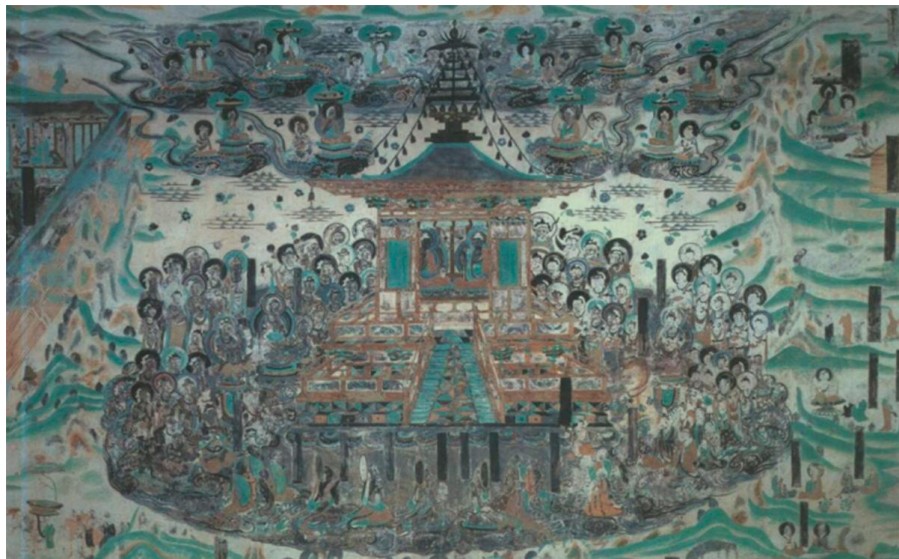

**Figure 4.** The "Ceremony in the Air", transformation of the *Lotus Sūtra* on the Southern Wall in Cave 23 during the flourishing Tang dynasty (704–780 CE) at Mogao Grottoes.

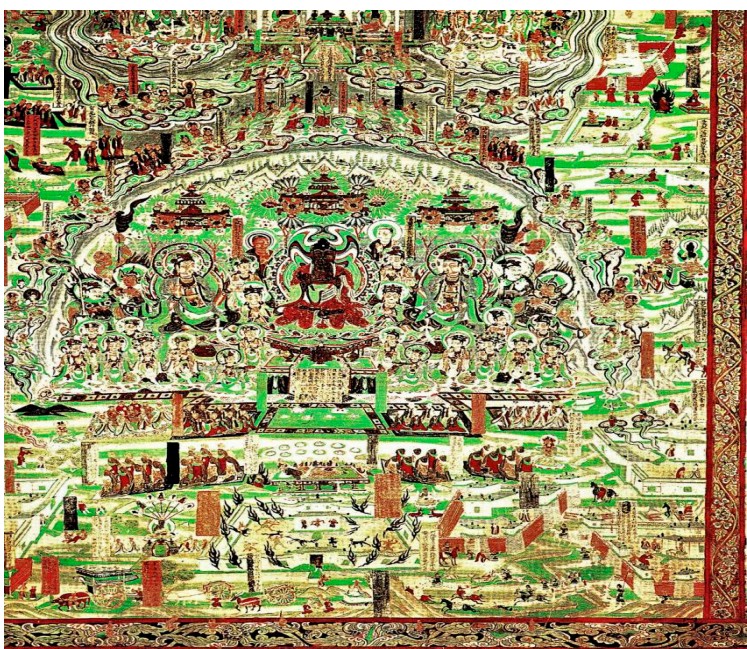

**Figure 5.** Cave 61 of the Mogao Grottoes during the Five Dynasties period (907–979 CE) and the depiction of the *Lotus Sūtra* transformation.

2.1.2. The Reflection of the "Dharmakāya" Concept by the "Two Adjacent-Seated Buddhas"

In the context of iconic combinations in Grottoes, it is not uncommon to find examples of the "Two Adjacent-Seated Buddhas" as standalone revered icons alongside others. So, what unique significance does the independent popularity of the "Two Adjacent-Seated Buddhas" as objects of veneration hold? Regarding the "Two Adjacent-Seated Buddhas", Daosheng 道生 (ca. 360–434 CE) provides the following explanation:

> By dividing the seat and having them sit together, it signifies that existence need not always be existent, and non-existence need not always be non-existent. The distinction between existence and non-existence arises from the realm of aggregates. Is this not a sacred concept? It also reflects the impermanent nature of Nirvāṇa, suggesting it won't last forever.[4]

It is clear that the "Two Adjacent-Seated Buddhas" is intended to illustrate the state of neither birth nor death, as well as the eternal life of the Buddha's Nirvāṇa. Jizang 吉藏 (529–623 CE) also stated, in his *Fahua yishu* 法華義疏 (Annotations on the Meaning of the *Lotus Sūtra*),

> The co-sitting of the Two Buddhas is a way to manifest Shakyamuni through Prabhūtaratna. While Prabhūtaratna's extinction is not truly an extinction, it symbolizes the absence of cessation. When Shakyamuni sings of the dual forests (indicating his Nirvāṇa), the meaning is likewise conveyed. Furthermore, Prabhūtaratna's extinction, though not an actual extinction, signifies non-extinction. This reveals that Shakyamuni neither arises nor ceases, yet he both arises and does not arise. The appearance of Prabhūtaratna serves to show that Shakyamuni truly transcends both birth and death, skillfully manifesting both arising and ceasing. Therefore, Shakyamuni co-sits to demonstrate Prabhūtaratna, with the purpose of teaching people to abandon the misconception of Shakyamuni's birth and death. This is why Prabhūtaratna is elevated to manifest Shakyamuni and to dispel the ailment of clinging to Shakyamuni's birth and death.[5]

The appearance of Prabhūtaratna signifies the past characteristic of his Nirvāṇa, where he became extinct yet not extinguished (Kanno 2008, p. 226). "The past Buddhas, Prabhūtaratna, and the present Buddha, Shakyamuni, sit together in the stūpa, signifying the

transcendence of time and space, as well as the Buddha's eternal and everlasting presence" (Yamada 2019, p. 18).[6] This representation illustrates that Shakyamuni Buddha is both born and not born, extinguished and not extinguished, symbolizing the eternal and pure nature of his Nirvāṇa and the purity of his Dharma body. As Jizang stated, in his *Fahua tonglue* 法華統略 (Summary of the *Lotus Sūtra*), the extinguishing and non-extinguishing of Shakyamuni and Prabhūtaratna signify the eternal nature of their lifespans in Nirvāṇa, culminating in the realization of the Dharma body, Vairochana Buddha:

> When Prabhūtaratna opens the stūpa with sound, it illuminates the closed stūpa, signifying the perpetual openness of the stūpa, which, in turn, reveals the eternal nature of Prabhūtaratna. The intention behind manifesting Prabhūtaratna's eternal nature is to demonstrate it through Shakyamuni, which is why he is seated alongside. Moreover, Prabhūtaratna's ability to produce sound represents his imperishable essence, leading to the realization of Vairochana. Prabhūtaratna's ascension to the sky implies his eternal dwelling in the realm of peace and extinguishment. Similarly, Shakyamuni's imperishable nature also leads to the attainment of Vairochana, and his ascent to the heavens signifies his abode in the realm of eternal tranquility and luminosity. With the ancient and contemporary Two Buddhas being as such, this applies to all Buddhas.[7]

In summary, the authors contend that when Shakyamuni and Prabhūtaratna manifest simultaneously within the Prabhūtaratna Stūpa, the interpretation of their unity transcends the individual Buddha nature of Shakyamuni or Prabhūtaratna. Instead, it represents a Buddha nature that is beyond all Buddhas,[8] serving as the Dharma body Buddha, i.e., Dharmakāya, the source of all Buddhas across time and space. In other words, the significance of the "Two Adjacent-Seated Buddhas" lies in "using the technique of montage to connect the past world of Prabhūtaratna Buddha with the present world of Shakyamuni Buddha in a single visual composition. By delving into the relationship between the past and present Buddhas, existence and non-existence, it profoundly expounds the uniqueness, eternity, and equality of the 'Dharma body.' This artistic portrayal of the Two Adjacent-Seated Buddhas employs an intuitive and sensory approach to make the abstract concept of the 'Dharma body' tangible and relatable, using representational art to elucidate the formless nature of the 'Dharma body'" (Lin 2012, p. 141).

### 2.2. The Relationship between the Parinirvāṇa Image, King Ashoka's Offering of Soil, and the Supreme Dharma Body

The upper section of the "Buddha Story Stele" centers around the "Two Adjacent-Seated Buddhas". Below, on the left, is the Parinirvāṇa image, while above are depictions of the story of King Ashoka offering soil and the Pensive Bodhisattva beneath the tree. In Buddhism, the concept of Parinirvāṇa carries dual significance. On the one hand, it is a part of the Buddha's life story, illustrating the events and conduct of Shakyamuni throughout his life. Simultaneously, the theme of Parinirvāṇa holds great importance in the beliefs of Mahayana Buddhism. As Kumārajīva noted in the *Da zhidu lun* 大智度論 (Mahāprajñāpāramitā Upadeśa, Treatise on the Great Perfection of Wisdom),

> As for the Dharma Nature, "Dharma" is referred to as Nirvāṇa, something that is indestructible and beyond dispute. "Nature" signifies their inherent characteristics, much like how there is a golden nature in yellow stone and a silver nature in white stone. Similarly, all phenomena in the world possess the nature of Nirvāṇa.[9]

Nirvāṇa embodies the essence of the Dharma, and "the concept of the 'Dharmakāya' originally stems from the personified and abstracted teachings attributed to the Buddha" (D. Wei 2008, p. 9). It is the manifestation of all phenomena, signifying the equivalence of Dharmakāya and Nirvāṇa. Therefore, within the philosophical framework of Mahayana Buddhism, Shakyamuni's Parinirvāṇa is endowed with the special significance of an eternal Dharmakāya nature, one that transcends birth and extinction. As explicitly stated by

Sengrui[10] in the *Yu yi lun* 喻疑論 (Discourse on Parables and Doubts), the Buddha's Dharmakāya is, indeed, Nirvāṇa:

> Even in ancient times, when the "Mahāparinirvāṇa Sūtra" had not yet surfaced, the Dharma Body Sūtra existed, clearly asserting that the Buddha's Dharma Body is synonymous with Nirvāṇa. The teachings of the past and the present, if they align and conform, are in accordance with the covenant.[11]

The depiction of the Nirvāṇa image[12] as part of the Buddha's life story represents the Theravada perspective on the cycle of life and death (Collins 1990, 2010; Gombrich 1988). The emergence of the "Two Adjacent-Seated Buddhas" enriches the Theravada concept of Nirvāṇa with the Mahayana idea of the eternal life of the Buddha and the perpetual presence of the Dharma body. This elevation transforms it into a spiritual state beyond birth and extinction.

As a result, the Nirvāṇa image on the upper right side of the "Buddha Stories Stele" carries dual significance, and it harmonizes doctrinally with the central image of "Two Buddhas Seated". Additionally, the story of King Ashoka's offering of soil as a causal (Figure 6) is closely related to the Nirvāṇa of Shakyamuni. According to the records in the *Xian yu jing* 賢愚經 (Sūtra of the Wise and Foolish),

> Once upon a time, the Buddha was in the city of Shrāvastī, in Jeta's Grove, at Anāthapindika's Monastery. At that time, the World-Honored One, along with Ananda, went into the city to beg for alms in the morning. They encountered a group of children playing on the road. Each of them had gathered soil, shaping it into palaces, storehouses, and treasure troves filled with grains. Among these children, there was one who, from a distance, spotted the approaching Buddha. Witnessing the radiant countenance of the Buddha, he felt a deep sense of reverence and inner joy. His heart brimmed with the desire to make an offering. He promptly took some grains from one of the storehouses and, with his hand, intended to offer them to the Buddha. However, due to his young age and small stature, he asked another child, "I will climb on your back to offer these grains". The other child readily agreed and said, "Certainly". The first child then climbed onto the second child's back, grains in hand, and approached the Buddha to make the offering. The Buddha accepted the grains into his alms bowl, lowered his head to receive the soil, and, once received, he handed it to Ananda, saying, "Take this and use it to plaster my dwelling" … The Buddha said to Ananda, "The child who made a joyful offering of the soil has sufficiently smeared one side of my dwelling with it. Because of this merit, after I attain Parinirvāṇa in one hundred years, that child will become a king by the name of Ashoka. The second child will become a great minister, ruling over all the lands of Jambudvīpa. They will promote the three jewels, establish extensive offerings, distribute relics, and construct eighty-four thousand stupas throughout Jambudvīpa for my sake".[13]

King Ashoka, during his childhood, once presented soil symbolizing grains as an offering to Shakyamuni Buddha. Consequently, Shakyamuni Buddha foretold to his attendant Ananda that this child would later become a chakravarti, an ideal universal ruler. After Shakyamuni Buddha's Parinirvāṇa, he would distribute the Buddha's relics throughout Jambudvīpa and construct eighty-four thousand stupas in his honor. Therefore, the narrative of King Ashoka's soil offering can be regarded as a post-Nirvāṇa story associated with the Buddha. The presence of the Nirvāṇa image and the story of King Ashoka's soil offering related to Nirvāṇa, surrounding the "Two Adjacent-Seated Buddhas", further emphasize the profound significance of Shakyamuni Buddha's eternal Dharmakāya.

### 2.3. Following the Nirvāṇa of Shakyamuni Buddha: Maitreya's Birth and Kaśyapa's Offering of His Robe

On the left side of the story depicting King Ashoka's offering of soil, there is an image of a Bodhisattva meditating beneath a tree. The depiction of the Pensive Bodhisattva is a

relatively common motif during the Northern and Southern dynasties (386–589 CE). It can be found in various locations, such as the Kizil Caves, the Mogao Grottoes, the Yungang Grottoes, and the Longmen Grottoes, each displaying its distinct regional characteristics. W. Wei (2017, pp. 383–431) has conducted a comprehensive regional analysis, which can be referred to for further insights.

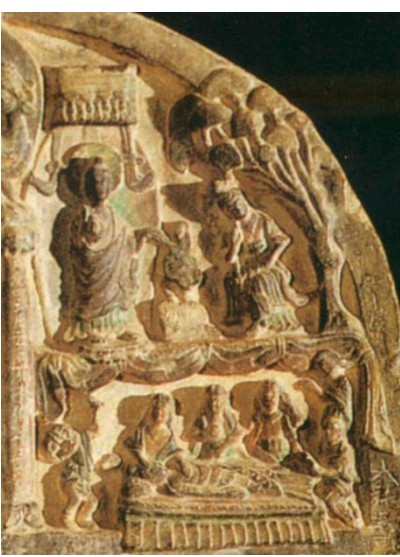

**Figure 6.** The upper right section of the "Buddha Story Stele" includes depictions of the Nirvāṇa image, the story of King Ashoka offering soil, and the Pensive Bodhisattva beneath the tree.

2.3.1. The Identity of Pensive Bodhisattva Icons and the Belief in Maitreya's Future Birth

Regarding the identity of Pensive Bodhisattva icons in the region of Maiji Mountain and their connection to the belief in Maitreya's future birth, previous scholars have presented four different viewpoints based on their combination with the Cross-Legged Bodhisattva.

(1) B. Zhang (1980, p. 342) and Haruo Yagi (2003, p. 61) believe that the Pensive Bodhisattva in Half-Lotus Position represents the image of Prince Siddhartha, while the Cross-Legged Bodhisattva symbolizes Maitreya Bodhisattva before attaining Buddhahood. (2) Kengo Toh (1998, p. 269) suggests that the Pensive Bodhisattva's image, besides being that of Maitreya Bodhisattva, can also be considered the Venerable One engaged in Bodhisattva practices, awaiting the right time. (3) X. Zhang (1990, p. 285), on the other hand, believes that the upper parts of the Cross-Legged and Pensive Bodhisattvas in the Maiji Mountain caves represent imagery of Bodhisattvas dwelling in the Pure Land. (4) Kazu Uehara (2006, pp. 22–23) posits that both the Cross-Legged and Pensive Bodhisattvas represent Maitreya Bodhisattva. After collecting extensive materials and analyzing and comparing them, W. Wei (2017, p. 431) concludes that "Based on additional evidence such as the Northern Liang stone pagoda, it can be confirmed that the combination in Maiji Mountain [referring to the pairing of the Cross-Legged Bodhisattva and Pensive Bodhisattva] represents different manifestations of Maitreya Bodhisattva. This indicates that during the Northern Wei period, there indeed existed the form of a Pensive Bodhisattva in the Half-Lotus Position, representing Maitreya Bodhisattva".

As previously discussed, the upper icons in the Northern Wei "Buddha Story Stele" from Cave 133 at Maiji Mountain illustrate the theme of Shakyamuni Buddha's Dharma body after his Parinirvāna. This theme is conveyed through depictions of Shakyamuni's Parinirvāna, the story of King Ashoka offering soil, and the representation of the "Two Adjacent-Seated Buddhas". There exists a profound connection between Maitreya and Shakyamuni's Parinirvāna, as Maitreya is portrayed as the future Lord of the Saha world, set to be born in this world 5.67 billion years after Shakyamuni's Parinirvāna, as documented in the "*Shi lao zhi* 釋老志 (Record of Buddhism and Daoism)" of the *Book of Wei*:

Before Shakyamuni Buddha, there were six Buddhas. Shakyamuni attained enlightenment during the current auspicious eon. The text foretells that Maitreya Buddha will appear in the future, continuing the lineage after Shakyamuni, and descend into this world.[14]

Taking all these factors into consideration, the authors suggest that the Bodhisattva depicted under the tree in the upper section of the "Buddha Story Stele" should be understood as Maitreya Bodhisattva. Both the Cross-Legged Maitreya and the Pensive Maitreya originate from the accounts found in the *Mile xiasheng jing* 彌勒下生經 (The Prophecy of Maitreya). The key distinction lies in the fact that the Cross-Legged Bodhisattva represents Maitreya's spiritual practice in the inner courtyard of Tuṣita Heaven before his descent, while the Pensive Bodhisattva symbolizes Maitreya's practice under the dragon flower tree after his descent into the human realm (RICBCC and MARI 2003, p. 314). Therefore, the presence of Maitreya Bodhisattva beneath the tree signifies his incarnation into the world of humans, marking the final stage of his Bodhisattva journey towards Buddhahood. At this point, Shakyamuni has already entered Nirvāṇa. Hence, the Pensive Maitreya Bodhisattva, the Nirvāṇa image, and the story of King Ashoka's soil offerings are thematically interconnected, forming part of the same narrative.

### 2.3.2. Interpreting the Story of Kāśyapa Offering His Robe

In regard to the icon on the left side of the "Two Adjacent-Seated Buddhas" (Figure 7), Y. Zhang (2017, p. 54) identifies it as a mountain discourse, while Qingxiang Chen (2004, pp. 132–33) and Jingjie Li (2008, p. 498) propose that it represents Prince Siddhartha's tonsure and renunciation. The term "mountain discourse" is a broad categorization and does not pertain to a specific Buddhist story, resembling more of a scene description. Conversely, the act of "tonsuring and renunciation" is indeed a significant element in Buddhist stories. However, according to the *Taizi ruiying benqi jing* 太子瑞應本起經 (The Sūtra on the Auspicious Birth of the Crown Prince), it is recorded as follows:

> As I journeyed deep into the mountains, reaching a secluded and tranquil spot, I came across a Bodhi tree, and the surroundings were pristine as far as the eye could see. I contemplated, "I have already left my home behind and sought refuge in these secluded mountains and wetlands. It would not be appropriate to adorn my hair in the manner of an ordinary person" … Celestial beings brought forth a razor, and my hair fell naturally, accepted by the heavens.[15]

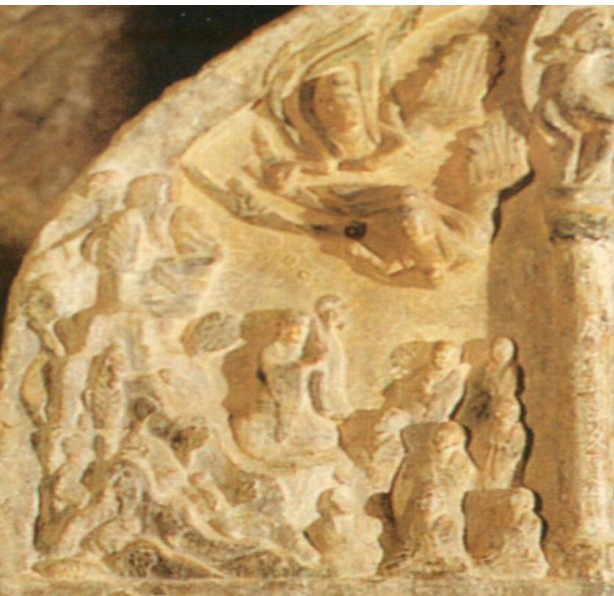

**Figure 7.** On the upper left side of the "Buddha Story Stele", there is a depiction of the story where Kāśyapa offers his robes to the Buddha.

This indicates that Prince Siddhārtha's ordination as a monk took place under a solitary Bodhi tree deep in the mountains, with a celestial being presenting him with a razor. In the image, the Bodhi tree is absent, and several figures are seen either sitting or standing in front of him. Therefore, the authors believe that this representation does not conform to the description of his ordination.

We observe that this story takes place in the mountains. In the image, a person is sitting cross-legged, and, in front of him, several people are either standing or sitting, paying their respectful homage. According to the *Mile xiasheng jing*, before his Parinirvāṇa, Shakyamuni entrusted Kāśyapa and others to await the coming of Maitreya, passing on his own robes and alms bowl to him (Figure 8). During the protracted period of waiting, Kāśyapa meditated on Jizu Mountain until the time of Maitreya's Buddhahood:

> At that time, the World-Honored One said to Kāśyapa, "… Kāśyapa, you should not enter Parinirvāṇa either; you must wait for Maitreya to appear in this world" … In the village of Vaisali, on the border of Magadha, Great Kāśyapa resided on that mountain. Then, the Tathāgata Maitreya, along with an innumerable assembly of thousands of people, came forward and circumambulated this mountain. Through the Buddha's benevolence, all the spirits and deities opened the door, allowing them to witness Kāśyapa's meditation cave … At that moment, Ananda, Maitreya Tathāgata should take the Sangharama robe that Kāśyapa is wearing. At that time, Kāśyapa's body disintegrated and scattered into stardust.[16]

Consequently, considering the image content and traditional accounts, the authors propose naming this icon "Kāśyapa Offering His Robe". Both "Kāśyapa Offering His Robe" and "King Asoka Offering the Soil" share similarities, as they can be viewed as subsequent stories to Shakyamuni Buddha's Parinirvāṇa. These narratives align with the themes depicted in the upper right part of the "Buddha Story Stele", which reflect Shakyamuni's Parinirvāṇa and the birth of Maitreya.

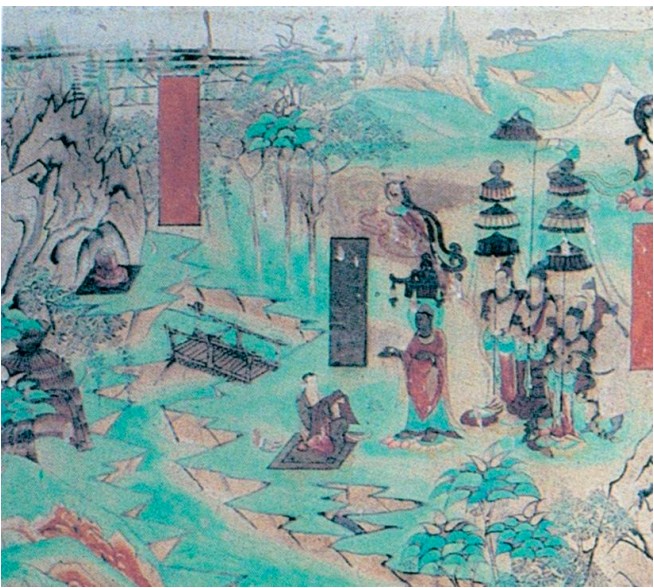

**Figure 8.** This scene is based on the narrative from Maitreya's Birth Sūtra, found in Cave 25 of the Yulin Grottoes, dating back to the Tang dynasty (618–907 CE).

In summary, the upper section of Cave 133 in the Maiji Mountain Grottoes conveys stories such as the Nirvāṇa scene, the story of King Asoka offering the soil, Maitreya's descent into the world and his meditation beneath the tree, and the episode of Kāśyapa offering his robe after Maitreya's Buddhahood. These stories together illustrate the phase that occurred after Shakyamuni Buddha's Parinirvāṇa. The central icon of the "Two Adjacent-

Seated Buddhas" perfectly complements this time frame by portraying the concept of the Dharma body, showcasing the form of the Buddha after Shakyamuni's Parinirvāna.

## 3. Depicting Shakyamuni's Final Bodhisattva Journey with a Focus on the Cross-Legged Bodhisattva

*3.1. The Middle Section: Depicting the Life and Stories of Bodhisattva Siddhārtha*

In the middle section of the "Buddha Story Stele", the four narrative panels surrounding the central figure of the seated Bodhisattva Siddhartha illustrate scenes from the life of Siddhārtha Gautama, as well as stories from his previous births (Figure 9). Among these panels, the images representing "Conception through an Elephant Dream",[17] "Birth under the Tree", and "Learned Youth Pave his Hair on the Ground" are relatively straightforward to identify. However, there is ongoing scholarly debate regarding the lower left section. Y. Zhang (2017, p. 54) identifies it as the story of Siddhartha subduing the six heretic teachers, and Wenbo Yang (2019, pp. 58–66) suggests that it portrays the story of Siddhārtha subduing a venomous dragon, while Dorothy C. Wong (2004, p. 127) considers it a story about Māra's assault.

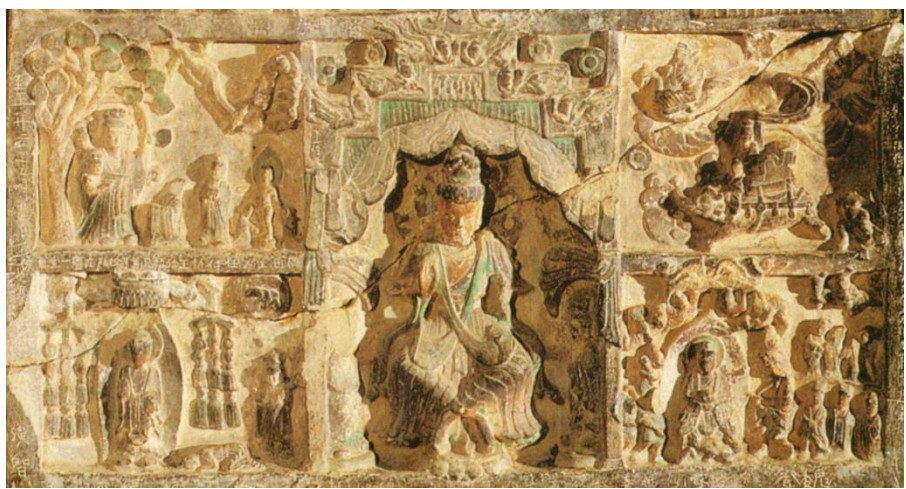

**Figure 9.** Depictions related to the Cross-Legged Bodhisattva as the central figure in the middle section of the "Buddha Story Stele".

3.1.1. Naming the Icon of the "Four Heavenly Kings Offering Alms Bowls"

Upon a closer examination of the icon's content, we can discern the central figure of Shakyamuni seated in meditation, making the Dhyana Mudra and holding an alms bowl. To his left, four heavenly beings stand, dressed in long robes with voluminous sleeves and adorned with celestial crowns. The foremost figure in the group is shown presenting a gift to Shakyamuni (Figure 10). As per "The Sūtra on the Auspicious Birth of the Crown Prince", during Shakyamuni's meditation practice, the Four Heavenly Kings sensed his thoughts and created four stone alms bowls from rocks taken from four different mountains. They offered these stone alms bowls to Shakyamuni, who, through his supernatural powers, merged them into one bowl. This is the renowned tale of the "Four Heavenly Kings Offering Alms Bowls":

> The Buddha remained in deep meditation for seven days without any movement. The guardian spirit of the tree considered, "The Buddha has just attained enlightenment and is meditating for seven days, yet there is no one offering him food. I shall find someone to make an offering to the Buddha" ... The Buddha thought about how, in ancient times, all Buddhas had compassionately accepted offerings of food from people and held an alms bowl. It wasn't appropriate for him to receive food in his hands like other ordinary monks. At that moment, the Four Heavenly Kings, sensing the Buddha's intention to use an alms bowl, extended their arms and arrived at Mount Isha. Just as the Buddha had contemplated,

alms bowls naturally emerged from within the rocks, pure and fragrant, devoid of any impurities. The Four Heavenly Kings each took one of these alms bowls … The Buddha, realizing that taking just one bowl wouldn't be enough, accepted all four alms bowls. He stacked them in his left hand and pressed them down with his right hand, merging them into a single alms bowl that appeared as one in all directions.[18]

Based on the traditional scriptures and a careful examination of the image's content, the authors believe that it is more reasonable to name this panel "The Four Heavenly Kings Offering Alms Bowls". This scene depicts the moment when Shakyamuni Buddha, seated in profound meditation under the Bodhi tree after several days of enlightenment, received alms bowls offered by the Four Heavenly Kings. The alms bowl holds significant importance in the life of the Buddha, signifying his acceptance of alms and offerings from sentient beings. Furthermore, the alms bowl played a crucial role in numerous miraculous events described in various Buddhist scriptures, establishing it as an indispensable tool for the Buddha's teaching activities (Li 2011, pp. 42–43). Thus, the Buddha accepting alms bowls from the Four Heavenly Kings symbolizes the completion of his meditation practice and the fulfillment of his final Bodhisattva path. From this point forward, he began his journey as a Buddha, using this alms bowl to collect food offerings and enlighten sentient beings.

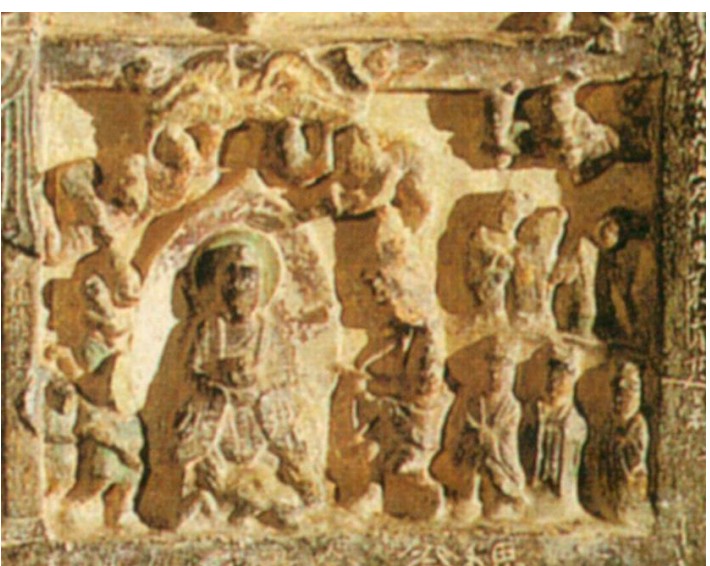

**Figure 10.** The "Four Heavenly Kings Offering Alms Bowls" icon on the Northern Wei "Buddha Story Stele" in Cave 133 of the Maiji Mountain Grottoes.

### 3.1.2. The Story of the "Learned Youth Pave His Hair on the Ground" and the Prophecy of Siddhartha Gautama's Enlightenment

The stories of "Conception through an Elephant Dream" and "Birth under the Tree" depict the beginning of Siddhārtha Gautama's final lifetime as a Bodhisattva, where he descended to Earth to complete his last Bodhisattva path. The "Four Heavenly Kings Offering Alms Bowls" marks the culmination of Siddhārtha's Bodhisattva path under the Bodhi tree, leading to his enlightenment and attainment of Buddhahood. Conversely, the stories of the "Learned Youth Pave his Hair on the Ground" and "Dīpaṃkara Buddha's Prophecy" relate to the prophecy and decision made by Siddhartha Gautama to achieve Buddhahood:

In the ancient days, during the time when Dīpaṃkara Buddha appeared in the world … I was a bodhisattva named Youth … Upon learning of a Buddha in the world, my heart rejoiced. I donned attire made of deerskin and set out to enter the city … Upon entering the city and encountering the people, they exhibited immense joy, smoothed the pathways, scattered fragrances, and lit incense. I in-

quired of a traveler, "What is the reason for these activities?" The traveler replied, "Today, a Buddha is expected to enter the city". The bodhisattva was overjoyed and thought to himself, "How marvelous! I shall now witness the Buddha and make my aspirations" … Before long, the Buddha arrived, and the king, his ministers, and the people all welcomed and paid homage, strewing various kinds of flowers. The flowers fell to the ground, but those that fell in front of the Buddha remained suspended in mid-air, creating a canopy above him. Later, two flowers took their place on the Buddha's shoulders. The Buddha discerned the bodhisattva's intention … and prophesied, saying, "In the future, after ninety-one aeons, during an era known as 'Virtuous', you will become a Buddha named Shakyamuni".[19]

As per the scriptures, during the era of Dīpaṃkara Buddha, when Shakyamuni was a young Bodhisattva still traversing the path, he was known as the "Youth Bodhisattva". When he encountered a Buddha during their journey, he smoothed the path for that Buddha, using his hair to pave the road. This act led to a prophecy from Dipankara Buddha that foretold his future Buddhahood as Shakyamuni. To consolidate, the stories within the central portion of the "Buddha Story Stele", surrounding the Cross-Legged Bodhisattva, all revolve around Shakyamuni's experiences, whether as a Bodhisattva in his final life completing his spiritual path, receiving offerings from celestial beings in the concluding stage of his journey, or a prophecy that, as the Youth Bodhisattva, after countless eons of being a Bodhisattva, he would inevitably attain Buddhahood. These stories collectively represent various facets of Shakyamuni's life, all stemming from his time as a Bodhisattva on the path.

*3.2. The Diversity of Identities within the Cross-Legged Bodhisattva: Maitreya Bodhisattva or Sumedha Bodhisattva?*

So, the central question now is the identity of the Cross-Legged Bodhisattva in the middle. Y. Zhang (2017, p. 54) suggests, "In the central niche on the middle tier, the seated Bodhisattva signifies Maitreya Bodhisattva within the Tuṣita Heaven". However, as discussed earlier, the stories within the central part of the "Buddha Story Stele" revolve around the experiences of Shakyamuni during his Bodhisattva path. These stories do not align with Maitreya, and it appears that pairing them together is evidently unreasonable.

X. Zhang (1990, p. 276), through statistical analysis, suggests that the term "Cross-Legged Bodhisattva" essentially refers to a Bodhisattva who resides within the Tuṣita Heaven, awaiting rebirth in the human realm, which is a characteristic of a Bodhisattva in the final stage of their spiritual journey. Therefore, among the many existing representations of the Cross-Legged Bodhisattva, a portion indeed represents Maitreya Bodhisattva. However, a significant part is likely to represent Shakyamuni before his enlightenment and other Bodhisattvas. Maitreya resided in the Tuṣita Heaven and preached to the celestial beings as a Bodhisattva in the final stage of his path (*Foshuo guan Mile pusa shang sheng dou shuai tian jing*, p. 418). Similarly, Shakyamuni, in his identity as Bodhisattva Sumedha, delivered teachings to the celestial beings in the Tuṣita Heaven before his final birth as a human being:

At that time, Bodhisattva Sumedha, having completed his meritorious practices and attained the tenth stage, in the life of a Bodhisattva, was nearing the possession of all forms of wisdom. He was reborn in the Tuṣita Heaven, known as the Holy One, Virtuous and Pure. Upon observing five significant events, he contemplated: "Considering King Shuddhodana's past karma, the couple who are truly worthy to be my parents; also, contemplating Queen Māyā's life, her lifespan is short, and she conceived the prince. She carried him for a full ten months, and after seven days of his birth, her life came to an end".[20]

Hence, considering that both Shakyamuni and Sumedha Bodhisattva were Bodhisattvas dwelling in the Tuṣita Heaven and teaching the Dharma to the celestial beings during their Bodhisattva phases, it becomes reasonable to depict Shakyamuni, in his Bodhisattva form as Sumedha, as the "Cross-Legged Bodhisattva".

The use of the "Cross-Legged Bodhisattva" to represent Shakyamuni's identity is not uncommon. For instance, the first niche at the third level on the right wall of the Guyangdong at the Longmen Grottoes in Luoyang, China, an icon of the Cross-Legged Bodhisattva, was crafted on 20th February in the second year of the Xiping era, during the reign of Yuanyou of Qi County, who held the position of governor of Jingzhou, in the Northern Wei dynasty (517 CE). The inscription reads, "Emperor Xuanzong's magnanimity and divine deeds are beyond the mundane world. The spiritual garden ascends to the realm of emptiness, transcending the dusty environment. If it were not to depict physical appearance to express the splendid grace, pursuing vocal teachings to exhibit the sublime path, how could it serve as the model for the images of the exalted master?"[21] The terms "physical appearance" and "exalted master" in this inscription likely refer to Shakyamuni Buddha (X. Zhang 1990, pp. 274–75). Another example can be found in Issue 18 of the Japanese magazine *Buddhist Art*, where a Cross-Legged Bodhisattva statue is featured, and the inscription on the base behind the statue reads, "□□□□ [indistinct words] Dedicated as a memorial by a son in mourning, a work of Shakyamuni, who continually gathers with the Three Jewels. First day of the seventh month in the second year of Yanchang".[22] Based on the inscription, it can be concluded that this Cross-Legged Bodhisattva represents Shakyamuni Buddha. In the 110th cave of the Kizil Caves in Xinjiang, China, there is an icon of a Cross-Legged Bodhisattva seated on a square seat. According to the research of Mingyi Ding (1983, p. 84), this Cross-Legged Bodhisattva is an image of Shakyamuni, who "once resided in the Tuṣita Heaven, preparing to descend to the world, and observed five significant events". Therefore, taking into consideration the content of the icons in the middle section of Cave 133 at Maiji Mountain, as well as the original meaning of the Cross-Legged Bodhisattva and references from other regions like the Longmen Grottoes and Kizil Caves, the authors believe that the Cross-Legged Bodhisattva in this context should depict the scene of Shakyamuni before his enlightenment, when he resided as following Bodhisattva in Tuṣita Heaven. The Bodhisattva's identity in this scene is likely to be that of Sumedha Bodhisattva.

In summary, within the central portion of Cave 133 at Maiji Mountain from the Northern Wei period, where the "Buddha Story Stele" is situated, the central focus is represented by the seated cross-legged figure of Shakyamuni as following Bodhisattva. In this context, Shakyamuni embodies his role as a Bodhisattva dwelling in the Tuṣita Heaven, prior to his descent into the human realm. Symmetrically distributed on both sides of this central figure are two sets of stories related to the Buddha's life and past lives. The two upper panels, "Conception through an Elephant Dream" and "Birth Under the Tree", form one set, portraying the inception of Shakyamuni's final Bodhisattva journey. The two lower panels, "Youth's Hair Offering" and "The Four Heavenly Kings Offering Alms Bowls", form another set, depicting Shakyamuni's determination to become a Buddha and the ultimate completion of his spiritual practice beneath the Bodhi tree while receiving offerings from celestial beings. These narratives distinctly convey the primary theme of the middle section of the "Buddha Story Stele", which revolves around the causes, processes, and outcomes of Shakyamuni's final Bodhisattva journey.

*3.3. Depicting Shakyamuni's Journey of Spreading the Teachings after His Enlightenment with a Focus on the Seated Preaching Buddha*

In the lower section, centered around the Buddha preaching in a seated posture, scenes on the left and right sides depict "The First Turning of the Wheel of Dharma at Sarnath"[23] and debates between Manjushrī and Vimalakīrti (Figure 11). "The First Turning of the Wheel of Dharma at Sarnath" refers to Shakyamuni's journey to Sarnath after his enlightenment under the Bodhi tree, where he met five ascetic companions with whom he had previously practiced austerities. He expounded his newly realized teachings to them, primarily covering the Four Noble Truths and the Eightfold Path of the Hinayana tradition (Ren 1985, p. 310):

At one time, the Buddha resided in the deer park of Sarnath, in the abode of sages. During that time, the World-Honored One addressed pañcavaggiyā, the name for the group of five ascetics, saying, "You should contemplate the Noble Truth of Suffering, a teaching you have not heard before … Moreover, reflect upon the wisdom related to the Noble Truth of Suffering, an aspect you have not been exposed to. Once again, when it comes to the cessation of suffering, recognize it and bear witness to the cessation, a doctrine previously unknown to you. Ponder it thoughtfully. Once more, monks, having recognized the Noble Truth of Suffering and witnessed it, contemplate what you haven't encountered earlier. Additionally, contemplate the Noble Truth of the Cessation of Suffering, which you have known, testified to, and emerged from, even though you hadn't heard of it previously.[24]

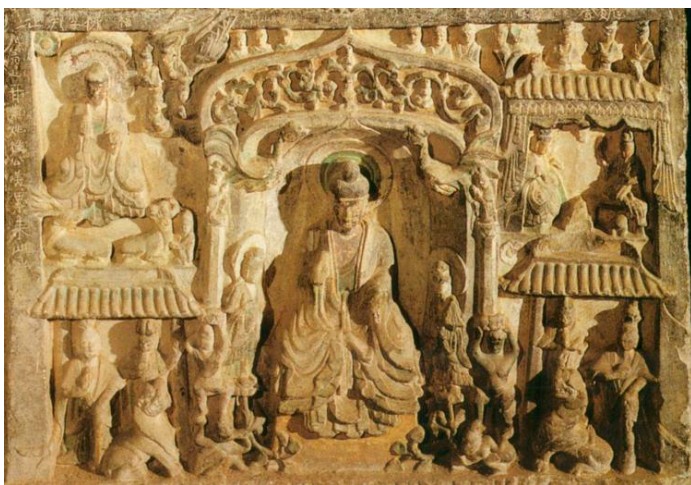

**Figure 11.** The lower section of the "Buddha Story Stele" showcases a seated Buddha in a teaching posture, scenes from the "First Turning of the Wheel of Dharma" at Sarnath, and Vimalakīrti's inquiry about illness.

The debate between Bodhisattva Manjushrī and Vimalakirti originates from the *Vimalakirti Sūtra*, specifically in the chapter titled "Manjushrī Asks about Illness". Vimalakirti, who had already become a Buddha countless eons ago, known as the Tathāgata "Jeweled Accumulation", was born in the city of Vaisali in the eastern realm of "Wonderful Joy" during Shakyamuni Buddha's lifetime. He took birth as a layman and assisted in the propagation of the Dharma alongside Shakyamuni Buddha (*Weimo jing xuan shu*, p. 532). At one point, Shakyamuni Buddha accepted the invitation of five hundred elder disciples to give a Dharma discourse in Vaisali. However, Vimalakīrti claimed to be ill and unable to attend. The Buddha wished to send his arhats and Bodhisattvas to visit Vimalakīrti, but his disciples were apprehensive about Vimalakīrti's exceptional debating skills and hesitated to go. Ultimately, it was the Bodhisattva of Wisdom, Manjushrī, who bravely volunteered to visit Vimalakīrti. In the ensuing dialogue, Vimalakīrti expounded the doctrine of the Middle Way, illustrating the nonduality of all phenomena. Therefore, the debate between Bodhisattva Manjushrī and Vimalakīrti symbolizes the central teachings found in the entire *Vimalakīrti Sūtra*.

Since the Wei and Jin dynasties (220–420 CE), the study of Buddhist theory has been largely centered around the Prajñāpāramitā teachings (Ren 1985, p. 460), and this Sūtra played a pivotal role in the propagation of Mahayana Prajna philosophy during this era (Ren 1985, p. 324). The monk Seng Zhao (384–414), in his treatise Seng Zhao's Discourse, unified Prajna and "Upāya" (skillful means), referring to them collectively as "Great Wisdom" (Ren 1985, pp. 476–77):

The true nature of all phenomena is referred to as Prajnaparamita, and the ability not to grasp onto appearances is the skill of Upāya. Transforming sentient beings

is known as Upāya, as it remains untarnished by worldly entanglements and carries the power of Prajnaparamita. Therefore, the gateway to Prajnaparamita focuses on emptiness, while the gateway to Upaya deals with existence. By addressing existence without falling into delusion about emptiness, one can continually engage with existence without becoming entangled. By not shunning existence and contemplating emptiness, one can observe emptiness without conceptualization.[25]

In his perspective, Prajnāpāramitā, known as "Great Wisdom", must encompass two essential aspects. Firstly, there is Prajnāpāramitā itself, also referred to as "Wisdom". Its purpose is to realize the true nature of all phenomena, achieving the state of "engagement with existence without entangled" and the "contemplation of emptiness without aversion to existence". Secondly, there is "Upāya", a concept associated with Buddhist "skillful means" or "Expedient" teachings. Its role is to "cater to the needs of sentient beings". Therefore, the union of "Upāya" and "Prajna" is termed "Expedient Wisdom". In the *Vimalakīrti Sūtra*, Seng Zhao emphasized that "Expedient Wisdom is pivotal within this sūtra" (*Zhu Weimoji jing*, p. 379). The profound wisdom of Buddhist doctrine and the skillful means to benefit sentient beings find their unity in the teachings of Vimalakīrti, highlighting the crucial role of "Expedient Wisdom" and "Prajna" within the context of Mahayana wisdom studies.

In summary, the authors' interpretation suggests that the lower section of Cave 133 in Maiji Mountain, known as the "Buddha Story Stele", is designed with a specific purpose. It features the central figure of a seated Buddha in a teaching posture, flanked by depictions of the Sarnath's "First Turning of the Wheel of Dharma", representing the Theravada teachings (the Four Noble Truths and the Eightfold Path), and images of Vimalakirti and Manjushrī, representing Mahayana Prajna teachings. This configuration, in a way, helps to elucidate the significance of the Buddha's teaching gesture, emphasizing the theme of Gautama Buddha's post-enlightenment phase, when he propagated both Theravada and Mahayana Buddhist doctrines.

## 4. Conclusions

This study delves into the relationship between the central icons and the surrounding narrative reliefs in "Buddha Story Stele" from Cave 133 of the Maiji Mountain. In contrast to previous research, this paper suggests that the three central sculptures—the "Two Seated Buddhas Side by Side", the "Cross-Legged Bodhisattva", and the "Buddha in Teaching Posture"—represent three distinct phases in the life of Gautama Buddha. These phases correspond to the Dharmakāya Buddha after Nirvāṇa, the Bodhisattva Gautama in the form of Sumedha Bodhisattva, and the Buddha Gautama, respectively. The basis for this interpretation lies in the understanding that, after achieving enlightenment, the Buddha disseminated his teachings in the human realm, which is symbolized in the lower section of the stele. Gautama, as a following Bodhisattva, resided in the Tuṣita Heaven above the earthly plane, and his presence is depicted in the middle section of the stele. Lastly, the Dharmakāya Buddha after Nirvāṇa transcended the physical realm and existed in the formless, colorless realm, above the realms of desire and form deities, as symbolized in the uppermost part of the stele.

The exploration of the "Buddha Story Stele" in the Maiji Mountain Grottoes involves not only deciphering the intricate artistic narrative within the stele but also delving into the historical and cultural context that surrounds it. This stele can be described as crafting a transcendent religious narrative that spans across time and space. It achieves this by depicting the various stages of Sakyamuni's life, complemented by the instructional endeavors of numerous Bodhisattvas and Buddhas. Moreover, through this unique narrative, not only do the Buddha's teachings, beliefs, wisdom, and art achieve the perfect fusion, but it also vividly presents the people's understanding of the Buddhist teachings at that time, the eulogy of the Buddhist faith.

This study has greatly enhanced our grasp of the intricate connection between Buddhist culture and the Maiji Mountain area during the Northern Wei dynasty. Buddhist culture transcends mere religious belief; it embodies a rich cultural legacy and innovation, aiming to assimilate the teachings of Buddha into the collective consciousness, motivating individuals to engage in virtuous actions and cultivate compassion. Therefore, by safeguarding and transmitting this invaluable cultural heritage, we not only enhance our understanding of history but also offer a wellspring of wisdom and compassion for contemporary society. This enables the ancient narrative to be effectively harnessed in today's context, enlightening minds and perpetuating the tradition of Buddhist wisdom.

Furthermore, through an examination of the relationship between the central sculptures and the surrounding narrative reliefs in the "Buddha Story Stele", this study confirms that the Buddhists in the Maiji Mountain region of China during the 5th to 6th centuries not only inherited the essence of original Indian Buddhist stories but also assimilated the creative ideas and themes of contemporary indigenous Buddhist art. They then embarked on the sculptural work of the stele, giving rise to the distinctive and captivating paradigm of the "Buddha Story Stele". As a result, this enriches our understanding of the religious art in the icons and steles of the Maiji Mountain.

**Author Contributions:** Conceptualization, Z.L. (Zejie Lin) and Z.L. (Zhijun Li); Formal analysis, M.X.; Writing—original draft, Z.L. (Zejie Lin) and Z.L. (Zhijun Li); Writing—review & editing, M.X. All authors have read and agreed to the published version of the manuscript.

**Funding:** This research was funded by Social Science Foundation of China, grant number 15AZJ003 and National Postdoctoral Fellowship Program, grant number GZC20230597.

**Institutional Review Board Statement:** Not applicable.

**Informed Consent Statement:** Not applicable.

**Data Availability Statement:** No new data were created or analyzed in this study. Data sharing is not applicable to this article.

**Conflicts of Interest:** The authors declare no conflict of interest.

## Notes

[1] As W. Wei (2017, p. 378) suggests, among the combinations of the Three Buddhas, there is one form composed of Sakyamuni Tathāgata (emphasizing the past-life Tathāgata), Sakyamuni (the present Tathāgata), and Maitreya (the future-life Tathāgata), or by combining Sakyamuni Tathāgata (past and present) and Maitreya (the future).

[2] Explore the teachings of the *Lotus Sūtra* and modern interpretations in the works of Lopez and Stone (2019). Their insightful guide delves into the essence of the classic, showcasing the pivotal role of commentary in revitalizing the ancient literature for contemporary understanding. The English translation of the *Lotus Sūtra* can be found in Reeves' (2008) book. The pages for Chapter 11 are from 235 to 246. Furthermore, Wang's (2005) scholarly investigation delves deeply into the significance of the *Lotus Sūtra* within East Asian Buddhism, elucidating its profound influence on China's vibrant visual culture. Wang's meticulous study offers a comprehensive understanding of the cultural impact of this revered text, highlighting its enduring significance in the region.

[3] Rhie (2010, p. 137) suggests that this represents an evolving iconography in China, with the appearance of Maitreya Bodhisattva alongside the Buddha pair in a unified depiction emerging around the first or second decade of the 5th century. It is noteworthy that the use of panels embedded in the thousand Buddha field became a prevalent form in Chinese Buddhist art from this period onward and is frequently associated with the *Lotus Sūtra*.

[4] *Miaofa lianhua jing shu*, T no. 1521, vol. 27, p. 13: "分半座,所以分半座共坐者,表亡不必亡,存不必存,存亡之異,出自群品,豈聖然耶,亦示泥洹不久相也". The authors translated this and the citations that follow.

[5] *Fahua yishu*, T no. 1721, vol. 34, p. 590: 所以二佛同坐者,正欲以多寶顯釋迦也。多寶滅既不滅,不滅示滅。釋迦雙林唱滅,義亦同然。又多寶滅既不滅,不滅示滅,即顯釋迦不生而生,生而不生。以多寶出現,欲顯釋迦實無生滅,方便生滅,故要釋迦共坐也。所以多寶欲顯釋迦者,正為稟釋迦教人,執釋迦實有生滅,故舉多寶以顯釋迦,破執釋迦生滅病也。

[6] Rhie (2010, p. 137) suggests that, in Chapter 11 of the *Lotus Sūtra*, the depiction of two Buddhas signifies the Mahāyānist concept of the distant past Buddha, Prabhūtaratna, returning in the present and future whenever the *Lotus Sūtra* is taught, illustrating the Ekayāna (One Vehicle) teaching. This portrayal emphasizes the immanence of Buddhas throughout time, challenging the notion of Nirvāṇa. Hanh (2009, pp. 103–4) also underscores the merging of the ultimate and the historical, emphasizing the capacity to encounter the ultimate in a specific moment within the tangible world. Nevertheless, the specific articulations and

interpretations diverge. Hanh accentuates that Prabhūtaratna embodies "the ultimate Buddha", while Shakyamuni epitomizes "the historical Buddha".

7　*Fahua tonglue*, T no. 0582, vol. 27, p. 511: 多寶開塔出聲,則閉塔明。常開塔出聲,故開塔明常,則多寶常義已顯。所以顯多寶常者,意在顯於釋迦,故要之就座。又多寶出聲,是多寶不滅,則成毗盧舍那。多寶升空者,住常寂滅光土。釋迦不滅,亦成毗盧舍那,升空亦住常寂光土。今古二佛既爾,一切諸佛類然。

8　D. Wei (2008, p. 10) pointed out that "the Dharma body is the source of all Buddhas, eternally existent, and not subject to human will … Considering the Dharma body as the source of all Buddhas in the past, present, and future, as the mother of all Buddhas, and affirming its eternal 'permanence', is a shared belief found in various Mahayana scriptures".

9　*Da zhidu lun*, T no. 1509, vol. 25, p. 298: 法性者,'法'名涅槃,不可壞,不可戲論之;'性'名為本分種,如黃石中有金性,白石中有銀性,如是一切世間法中皆有涅槃性。

10　Sengrui 僧叡 was a renowned Chinese Buddhist monk from the late 4th and early 5th centuries. He collaborated with Sengzhao and others in the revision of Buddhist scriptures translated by Kumārajīva, making him one of Kumārajīva's four great disciples.

11　*Chu Sanzang ji ji*, T no. 2145, vol. 55, p. 41: 什公時雖未有《大般泥洹》文,已有法身經,明佛法身即是泥洹,與今所出,若合符契約。

12　Lee (2010, pp. 47–48) notes that Nirvana images in cave temples of the sixth century are typically integrated into an overall pictorial program, symbolizing the infinite Buddhas who coexist across various times and places in the entire universe.

13　*Xian yu jing*, T no. 0202, vol. 4, p. 368: 一時佛在舍衛國祇樹給孤獨園。爾時世尊,晨與阿難,入城乞食。見群小兒於道中戲,各聚地土,用作宮舍,及作倉藏財寶五穀。有一小兒,遙見佛來,見佛光相,敬心內發,歡喜踴躍,生佈施心,即取倉中名為穀者,即以手掬,欲用施佛。身小不逮,語一小兒:"我登汝上,以穀佈施。"小兒歡喜,報言:"可爾。"即躡肩上,以土奉佛。佛即下鉢,低頭受土,受之已訖,授與阿難,語言:"持此塗汙我房。"……佛告阿難:"向者小兒,歡喜施土,土足塗汙佛房一遍,緣斯功德,我般涅槃百歲之後,當作國王,字阿輸迦。其次小兒,當作大臣,共領閻浮提一切國土,興顯三寶,廣設供養,分佈舍利,遍閻浮提,當為我起八萬四千塔"。

14　*Book of Wei*, vol. 114, p. 3029: 釋迦前有六佛。釋迦繼六佛而成道,處今賢劫。文中將來有彌勒佛,方繼釋迦而降世。

15　*Taizi ruiying benqi jing*, T no. 185, vol. 3, p. 467: 既歷深山,到幽閒處,見貝多樹,四望清淨。自念:"我已棄家,在此山澤,不宜複飾發如凡人意。"……天神奉剃刀,鬚髮自墮,天受而去。

16　*Foshuo Mile xiasheng jing*, T no. 0453, vol. 14, p. 422: 爾時,世尊告迦葉曰:"……大迦葉亦不應般涅槃,要須彌勒出現世間。"……摩竭國界毘提村中,大迦葉於彼山中住。又彌勒如來將無數千人眾,前後圍遶往至此山中,遂蒙佛恩,諸鬼神當與開門,使見迦葉禪窟……爾時,阿難!彌勒如來當取迦葉僧伽梨著之,是時,迦葉身體奄然星散。

17　In the *Sūtra on Past and Present Causes and Effects (Guoqu xianzai yinguo jing*, T no. 189, vol. 3, p. 626), it is recorded, "At that time, King Shuddhodana, with folded hands and palms together, paid respects to the heavenly gods. He carried the Crown Prince in front and placed him on the palanquin with seven Treasures … they entered the city together". Xiaofeng Sun (2021, p. 176), through a detailed examination of the relief image, discovered that, on the grand and splendid palanquin, there were two figures, one larger and one smaller. Therefore, it should be "interpreted as 'riding on the elephant back to the palace', with the smaller figure likely representing the recently born Prince Siddhartha and the larger one representing King Shuddhodana or Queen Maya". Whether it is "riding on the elephant during birth" or "riding on the elephant back to the palace", both depict stories from the final life of Bodhisattva Siddhārtha. This interpretation does not affect the arguments presented in this paper.

18　*Taizi ruiying benqi jing*, T no. 185, vol. 3, p. 469: 佛定意七日,不動不搖。樹神念:"佛新得道,快坐七日,未有獻食者,我當求人令飯佛。"……佛念先古諸佛哀受人施法皆持鉢,不宜如餘道人手受食也。時四天王,即遙知佛當用鉢,如人屈申臂頃,俱到頞那山上;如意所念,石中自然出四鉢,香淨潔無穢。四天王各取一鉢……佛念取一鉢不快餘三意,便悉受四鉢,累置左手中,右手按之,合成一鉢,令四際現。

19　*Taizi ruiying benqi jing*, T no. 185, vol. 3, pp. 472–73: 至於昔者,定光佛興世……時我為菩薩,名曰儒童……聞世有佛,心獨喜歡,披鹿皮衣,行欲入國……入城見民,欣然忽忽,平治道路,灑掃燒香。即問行者:"用何等故?"行人答曰:"今日佛當來入城。"菩薩大喜,自念:"甚快!今得見佛,當求我願。"……須臾佛到,國王臣民,皆迎拜謁,各散名華,華悉墮地。菩薩得見佛,散五莖華,皆止空中,當佛上如根生,無墮地者。後散二華,又挾住佛兩肩上。佛知至意……因記之曰:"汝自是後,九十一劫,劫號為賢,汝當作佛,名釋迦文"。

20　*Guoqu xianzai yinguo jing*, T no. 189, vol. 3, p. 623: 爾時善慧菩薩,功行滿足,位登十地,在一生補處,近一切種智,生兜率天,名聖善白……觀五事已,即自思惟:"……觀白淨王過去因緣,夫妻真正堪為父母;又觀摩耶夫人,壽命脩短,懷抱太子,滿足十月,太子便生,生七日已,其母命終"。

21　The original text is: 夫玄宗沖邈,跡遠於塵關,靈苑崇虛,理絕於埃境,若不圖色相,以表光儀,尋聲教以陳妙軌,將何以依希主像,仿佛神功者哉。

22　X. Zhang (1990), p. 27: □□□□□見為亡考造釋迦文一區作功,常與三寶共會。延昌二年七月十九日朔。

23　Sarnath is believed to be the place where Gautama Buddha, after attaining enlightenment in Bodh Gaya, went to find his previous five companions. It is where he preached the Four Noble Truths and established the Sangha, the monastic community. As a result, Sarnath, along with Lumbini, Bodh Gaya, and Kushinagar, is considered one of the four major Buddhist pilgrimage sites.

24　*Za Ahan jing*, T no. 0099, vol. 2, p. 103: 一時,佛住波羅㮈國仙人住處。爾時,世尊告五比丘:「此苦聖諦,本所未曾聞法,當正思惟……複次,苦聖諦智當複知,本所未聞法,當正思惟……複次,苦集滅,此苦滅聖諦已知當知作證,本所未聞法,當正思惟……複次,比丘!此苦聖諦已知,知已出,所未聞法,當正思惟……複次,苦滅聖諦已知、已作證出,所未聞法,當正思。

25　*Zhaolun*, T no. 1858, vol. 45, p. 150: 諸法實相,謂之般若;能不形證,漚和功也。適化眾生,謂之漚和;不染塵累,般若力也。然則般若之門觀空,漚和之門涉有。涉有未始迷虛,故常處有而不染;不厭有而觀空,故觀空而不證。

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
