# Peer review of "Narrative Integration: An In-Depth Exploration of the “Buddha Story Stele” in the Maiji Mountain Grottoes"

_religions, doi:10.3390/rel15030254_

Round 1
Reviewer 1 Report
Comments and Suggestions for Authors|
This study focuses on the iconology and iconography in Cave 133 of Maiji Mountain, with a particular emphasis on the careful identification of its identity as the Two Adjacent-Seated Buddhas and the Cross-legged Bodhisattva, exploring the symbolic significance behind its structural composition. This study effectively breaks through the static iconography research paradigm, but pays attention to the image program composed of different image units of the "Buddha Story" stele, and meticulously examines its upper, middle, and lower part, effectively distinguishing the controversial points in previous research. However, it is regrettable that this study focuses too much on individual research subjects themselves, and almost all of the content has not deviated from this stele. As Goethe once said:" He who knows one knows none." If we cannot considerably compare the expression of Shakyamuni's related image themes in different statues during the same period, and explore the range of this image on a larger scale, it will hide the importance of this research. When discussing the relevant images of the Lotus Sutra, the author's neglect of Eugene Wang's monograph is regrettable, reflecting a significant lack of understanding of the author's research on the English-speaking world. Additionally, in terms of the expression of the images in the upper, middle, and lower layers of this stele, the other way of understanding is to consider introducing the concepts of large time, medium time, and small time, viewing the two Buddhas sitting together as a large time that is close to eternity, viewing the middle layer, including multiple Jataka and stories of Shakyamuni, as medium time, and viewing its dissemination of Theravada Buddhism and Mahayana Buddhism as small time. |
Overall, the Buddhist terminology used, including English and Sanskrit writing, is very accurate, reflecting the author's high level of proficiency in English training and expression in Buddhist studies and Buddhist art history research. However, there are some problems of English translation in this paper. For instance:
1) Line 106-107: "dating to the first year of the Hongyuan era in the Western Qin dynasty (420 CE)". In this period, the Western Qin was a local regime with a relatively limited territorial scope in the early 5th century. For this reason, I prefer to use the Western Qin regime or kingdom, but not the Western Qin Dynasty. And the Hongyuan era is inaccurate, the right one is Jianhong建弘.
2) Line 334: "Records on Shakyamuni and Laozi" is 释老志 in Chinese. But the contents of this part in Weishu魏书 is not only focus on this two figures, but Buddhism and Daoism. In Chinese context, so-called Shi and Lao are refer to Buddhism and Daoism. Therefore, the accurate translation of 释老志 should be the record of Buddhism and Daoism.
3) Line 387, "Maitreyavyākaraa" is 弥勒下生经. Italic font should be used. And if the contents of this sutra is the same as the previous co-called Mile xiasheng jing (The Prophecy of Maitreya), why the author prefer to use sankrit here and use chinese pinyin and English translation in the previous discussions (Line 344-345).
Author Response
Dear Reviewer,
Thank you for your insightful feedback on our research. Your detailed review has been immensely helpful and has provided valuable insights for our study.
Firstly, we appreciate your recognition of our efforts to push beyond the traditional confines of static iconography research. Our aim is to delve deeply into the imagery of Cave 133 of Maiji Mountain, particularly focusing on the identification of the Two Adjacent-Seated Buddhas and the Cross-legged Bodhisattva, and exploring the symbolic significance behind their structural composition. Your acknowledgment of our analysis of the "Buddha Story" stele is encouraging, as it helps to address past research controversies effectively.
Secondly, we are grateful for your constructive criticism regarding our discussion of imagery related to the Lotus Sūtra. We have taken your advice to heart and have diligently consulted Eugene Wang's monograph, integrating it into our research framework. For instance, we referenced Wang's (2005) scholarly investigation to underscore the significance of the Lotus Sūtra within East Asian Buddhism. These additions enrich our research and provide a more comprehensive understanding.
Furthermore, we have addressed the English translation issues you raised by conducting thorough checks and revisions to ensure accuracy and consistency. Your feedback on this matter is greatly appreciated, and we are committed to maintaining precision and uniformity in language in our future writing endeavors.
Lastly, we understand and appreciate your suggestion regarding the scope of our study. While our primary focus remains on the dynamic composition aspects of the "Buddha Story Stele" in Cave 133, we acknowledge the potential for comparative analysis with other Buddhist images from the same period. We agree that exploring themes related to Shakyamuni across different statues could deepen our understanding. Incorporating the concepts of large time, medium time, and small time, as you suggested, could further enrich our perspectives.
Once again, thank you for your thorough review and guidance. Your insights will undoubtedly contribute to the refinement and impact of our research in the academic community.
Sincerely,
Reviewer 2 Report
Comments and Suggestions for Authors
I have no particular suggestions for Author/s; the work is extremely well thought out and convincing (although the illustrations leave something to be desired in terms of quality). Much of the bibliography is in Chinese: I would ask the Author/s to check for other recent publications not in Chinese, which have not been considered here.
I would suggest publishing the stele illustrations in a more readable format, and also request better quality photos of the pictorial comparisons (Mogao, Yulin, etc.) from the Author/s.
Also, the credits/illustrations are missing (if Author/s photo should be indicated, otherwise give the source and permission as well as the author's name, etc.).
Author Response
Dear Reviewer,
We deeply appreciate your valuable insights and recommendations regarding our research.
First and foremost, concerning the issue of images, we would like to emphasize that all the images referenced in our paper were personally captured by the authors during visits to the Maiji Mountain Grottoes. Thus, there are no copyright concerns associated with them.
Secondly, regarding the arrangement of the stele illustrations, we intentionally selected illustrations specifically related to the "Buddha Story Stele." We were cautious about comparing them with images from other grottoes such as Mogao and Yulin, as it might distract attention from the main research theme. However, we sincerely appreciate your suggestion, as it provides valuable guidance for future research in terms of image arrangement and comparison, which we will thoroughly consider in our upcoming work.
Lastly, we have enriched the bibliography by incorporating non-Chinese references, thereby ensuring a more balanced distribution of literature and facilitating a more extensive dialogue and exchange with existing research.
Once again, thank you for your thorough review and guidance. We will carefully consider your suggestions and strive for continuous improvement in our research.
Sincerely,